# Genotypic Variation in Nitrogen Use-Efficiency Traits of 28 Tobacco Genotypes

**André B. Andrade** [1],*[ID]**, Douglas R. Guelfi** [1][ID]**, Valdemar Faquin** [1]**, Fabrício S. Coelho** [2]**, Carolina S. de C. Souza** [1]**, Giulianno P. Faquin** [1]**, Kamila R. D. Souza** [3] **and Wantuir F. T. Chagas** [1]

1   Soil Science Department, Federal University of Lavras, 37200-900 Lavras/MG, Brazil; douglasguelfi@ufla.br (D.R.G.); vafaquin@ufla.br (V.F.); carolina_silva_@hotmail.com (C.S.d.C.S.); giuliannopf@gmail.com (G.P.F.); wantuirfilipe@gmail.com (W.F.T.C.)
2   Global Leaf Science & Research, British American Tobacco, 94970-470 Cachoeirinha/RS, Brazil; fabricio.coelho@bat.com
3   Institute of Natural Sciences, Federal University of Alfenas, 37130-001 Alfenas/MG, Brazil; krdazio@hotmail.com
*   Correspondence: andre.batp@hotmail.com

**Abstract:** Knowing the nitrogen use efficiency (NUE) of crops is crucial to minimize environmental pollution, although NUE is rarely provided for numerous genotypes in the tobacco (*Nicotiana tabacum* L.) crop. Through the growth of contrasting genotypes in nutritive solutions, we aimed to characterize five NUE components of 28 genotypes and to classify them according to their efficiency and responsiveness to nitrogen (N) availability. On average, physiological N use efficiency, N harvest index, and N uptake efficiency decreased by 16%, 4%, and 57%, respectively, under N-deficient conditions, while N utilization efficiency decreased by 43% at adequate N supply. The relative efficiency of N use varied from 35% to 59% among genotypes. All genotypes of the Virginia and Maryland varietal groups were efficient, and those of the Burley, Comum, and Dark groups were inefficient, while the responsiveness varied among genotypes within varietal groups, except for Maryland genotypes. Our findings are helpful in indicating genotypes with distinguished efficiency and responsiveness to N supply, which can be further chosen according to soil N level or affordability to N fertilizers worldwide in tobacco crops. In a general framework, this can lead to a more sustainable use of N and can support tobacco breeding programs for NUE.

**Keywords:** nitrogen use efficiency; screening; varietal groups; *Nicotiana tabacum* L., responsiveness; classification; grouping; N availability

## 1. Introduction

Nitrogen use-efficiency (NUE) improvement for crops is a very important issue to decrease soil, air, and water pollution and, at same time, increase the profitability of landholders through the rational use of mineral nitrogen (N) fertilizers. According to Kant et al. [1], an increase of 1% in NUE could save about 1.1 billion dollars per year. There are different ways to calculate the NUE of a given crop, being dependent on the harvest product of the crop (e.g., leaves, grains, etc.) and in what the researcher is interested to know [2]. The basic concept in our mind is that a crop with high NUE produces more with less.

NUE varies among plant species and, within a certain plant species, it also differs among the varieties that exist; in other words, NUE varies both inter- and intra-species. This variation of NUE is thought to be important as it allows the establishment of different varieties of the same species at different environmental conditions, i.e., different soil N levels. Hence, this implies that plants have

different internal mechanisms related to their NUE, or their embedded genetics for simplicity. The first step—and likely the easiest way—to increase the NUE of crops would be to both know and use in the best way that genetic variability available within a given plant species [3]. However, doing so is a difficult task, as the number of varieties can be large.

In the literature, surveys of NUE traits in different varieties of a given plant species are provided; for example, in sugarcane [4,5], potato [6,7], and rice [8]. However, in most cases, the number of varieties investigated is quite limited. This is understandable due to the time consumption in the conduction of many plants and the laboratory analysis required later. However, a better approach would be to try to embrace the highest number of genotypes as possible, allowing us to know the NUE specifically related to each genotype. This is only possible if the analysis involved in such studies is of low cost and is not time consuming. Thus, this could comprise a greater number of genotypes spanning a wide range of mechanisms related to NUE traits [9].

For this purpose, pot experiments represent a valuable tool for assessment, as the conduction of many genotypes under field conditions is unfeasible, beyond the existing soil variability. The study of genotypes under controlled conditions is faster, allows more genotypes to be investigated, is independent of pathogens and weather [6], and has minimal environmental variability. In addition, when the growing media is a nutritive solution, the evaluation of roots—responsible for nutrient uptake—becomes more reliable, rather than in experiments using pots filled with substrates [10]. Then, after screening for characterization of the NUE by plants, the choice of suitable genotype according to its N demand could be performed according to the area to be cultivated. In this way, farmers would have higher genotype diversity according to soil management conditions, being able to obtain acceptable yields with lower N inputs. Besides this, the knowledge of NUE of different varieties may also provide helpful data for tobacco breeding.

N fertilization in tobacco crops varies widely depending on the genotype. It is known, for example, that the varietal group Burley demands more N than others groups [11], but has similar yields. There are works that show that the difference of N fertilization is even 4-fold higher for Burley compared to other genotypes [12]. For tobacco genotypes, few works have used numerous genotypes in the study of NUE traits [13,14]. Thus, studies of the mechanisms behind NUE traits are limited to a few tobacco genotypes [15,16]. Again, this is comprehensive, since the growing process of many genotypes is time-intensive in terms of labor, and plant analysis is frequently time consuming with associated high costs (e.g., physiological analysis, $^{15}$N isotypes). However, a reliable selection of genotypes that differ in their NUE through cheaper and rapid assessment of plants would be interesting prior to the use of valuable sophisticated analysis, but it is expensive and unfeasible to do so in numerous genotypes.

For tobacco crop, N management is of particular interest as this nutrient plays an important role in the formation of nitrogenous compounds, namely, tobacco-specific nitrosamines (TSNAs), consequently linked to the quality of tobacco products. These compounds are carcinogens, found in negligible contents in fresh leaves but increases in the curing process [17]. TSNAs are formed by the reaction of nitrite (which is produced from nitrate ($NO_3$) uptaken from the soil) with alkaloids [18]. However, high $NO_3$ contents in the leaves are undesirable. In a general framework, the future tobacco market may require healthier products that comply with health regulatory agencies worldwide. One possible pathway is through the development of more efficient plants in the use of N through the breeding process.

In this sense, a species screening study represents a valuable tool for future studies in a breeding program. Accessing contrasting genotypes regarding NUE will make it possible to know the genes controlling traits related to N use in tobacco, such as chlorophyll and $NO_3$ content or N metabolism enzymes. For example, two genes were recently identified controlling NUE in tobacco [19]. Thus, the knowledge about the genes related to NUE may contribute to breeding programs aiming to obtain more efficient genotypes that are productive and that contain lower undesirable N compounds.

In light of the scarcity on works assessing a wide range of tobacco genotypes, the aims of this study were (1) to characterize the NUE traits of different *Nicotiana tabacum* L. genotypes when subjected

to both a deficient and an adequate N supply, and (2) to classify the genotypes according to their efficiency and responsiveness to N.

## 2. Materials and Methods

### 2.1. Tobacco Genotypes and Growth Conditions

A total of 28 tobacco (*Nicotiana tabacum* L.) genotypes were selected from the germplasm bank of the Product Center Americas of Souza Cruz company at Cachoeirinha, Rio Grande do Sul state, Brazil. The cultivars chosen covered five different varietal groups: Virginia, Burley, Comum, Dark, and Maryland (Table S1). The experiment was carried out under greenhouse conditions in Lavras, Minas Gerais state, Brazil, from 20 October to 12 December 2018. Seeds were sowed in seeding trays filled with organic substrate Tabaco−1® (MecPlant, Telêmaco Borba/PR), and then conducted in a floating system with deionized water. Then, 9 days after sowing (DAS), the floating system started to be fertilized with 10 mL $L^{-1}$ of a nutritive solution with the following concentration: 20% of N and $K_2O$, 10% $P_2O_5$, 0.15% Mg, 0.05% Fe, 0.025% Mn and Zn, 0.0125% B and Cu, and 0.005% Mo. At 25 DAS, tobacco seedlings were transferred to 30 L plastic trays filled with nutritive solution #1 of Hoagland and Arnon [20] at 20% and 50% of ionic strength (I.S.) for macro- and micronutrients, respectively. At 29 DAS, the nutritive solution was replaced by 40% and 50% of I.S. for macro- and micronutrients, respectively. The nutritive solution was permanently aerated throughout the period of plants adaptation to the hydroponic condition.

After the period of plant adaptation to the nutritive solution, plants of homogeneous size and vigor were selected to be transplanted to pots (3 L, one seedling per pot) with treatments at 31 DAS. The treatment nutritive solutions consisted of two contrasting N concentrations: deficient (2 mM) and sufficient (10 mM), which were established after previous tests with genotypes belonging to the Burley and Virginia groups in concentrations of N ranging from 1 to 12 mM. The concentration of the remaining nutrients was set at 60% and 100% of I.S. for macro- and micronutrients, respectively, based on the concentration of Hoagland and Arnon [20] solution #1 (Table S2). The solutions were renewed at 7, 14, and 18 days after transplanting (DAT), constantly aerated, and the pH adjusted to ~6.0 once per week. The experimental design was completely randomized, in a factorial scheme 28 (genotypes) × 2 (N concentrations, 2 and 10 mM), with four repetitions.

### 2.2. Plant Analysis and N Use-Efficiency Indexes

Plants were collected at 22 DAT (53 DAS) and roots were rinsed in deionized water. Then, plants were left in the greenhouse for pre-drying over 6 days. After this, plants were separated into roots, stem, and leaves, and dried in an oven with forced air circulation at 65 °C up to constant weight. The dry mass (DM) of each organ was recorded prior to grinding in a Wiley mill, and ground samples were analyzed for N concentration [21]. A standard reference material (NIST® SRM® 1573 a tomato leaves) was used in the digestion process to verify the accuracy of the method. N accumulation for each organ was calculated by the product of N concentration and DM.

Five indexes related to N use efficiency were calculated. N use efficiency (NUE; g DM $g^{-1}$ of N) represents the amount of plant DM produced per unit of N accumulated in the plant. Physiological N use efficiency (PNUE; $g^2$ DM $mg^{-1}$ of N) is the DM produced per N concentration. N harvest index (NHI; %) represents the amount of N in the leaves in relation to N in the whole plant [22]. N uptake efficiency (NUpE; mg N $g^{-1}$ roots) refers to the amount of N in the plant per root mass. Finally, relative N use efficiency (RNUE; %) represents the ratio of DM produced by the plants grown in N-deficient and N-sufficient supplies. The equations of the NUE indexes used in this study are represented below:

$$\text{NUE (g g}^{-1}\text{)} = \text{Plant DM (g)/N accumulation (g)} \tag{1}$$

$$\text{PNUE (g}^2\text{ DM mg}^{-1}\text{ of N)} = \text{Plant DM (g)/N concentration (mg g}^{-1}\text{)} \tag{2}$$

$$\text{NHI (\%)} = \text{(N accumulation in leaves/N accumulation in the plant)} \times 100 \quad (3)$$

$$\text{NUpE (mg N g}^{-1} \text{ roots)} = \text{N accumulation in plant (mg)/root DM (g)} \quad (4)$$

$$\text{RNUE (\%)} = [\text{Plant DM}_{\text{deficiency}} \text{ (g)/Plant DM}_{\text{adequate}} \text{ (g)}] \times 100 \quad (5)$$

To separate genotypes according to their efficiency and responsiveness to N, we calculated the N use efficiency (NUEf; g DM g$^{-1}$ N) based on the plant DM and N accumulation of plants grown under deficient and adequate N supplies as follows:

$$\text{NUEf (g g}^{-1}) = [\text{Plant DM }_{\text{adeq.}} \text{ - Plant DM }_{\text{def.}}]/[\text{N accumulation }_{\text{adeq.}} \text{ - N accumulation }_{\text{def.}}] \quad (6)$$

Data underwent analysis of variance (ANOVA) via the Sisvar software version 5.7 [23] and means were compared by Scott–Knott test at $p < 0.05$.

## 3. Results

### 3.1. Visual Symptoms of N Deficiency, Dry Mass (DM) Production, N Contents, and N Accumulation in Tobacco Genotypes

Plants grown in N-deficient solution revealed typical symptoms of this nutrient deficiency, that is, older yellowish leaves, while younger leaves showed a slightly lighter green color. A general view of the whole plants conducted in an N-sufficient solution revealed darker green plants, instead of a paler green color visualized in plants grown in a deficient solution, regardless of the genotype assessed (Figure S1). Higher plants were observed under adequate N supply, which was reflected in higher dry mass (DM) of roots, stems, and leaves under this growth condition (Table 1). The order of DM production followed stems < roots < leaves for N-deficient treatment, regardless of the genotype, and roots < stems < leaves on average when plants were grown under adequate N supply. Accordingly, higher root/shoot ratios were observed in plants conducted in an N-deficient solution. The average reduction of DM production when plants were grown under N deficiency was 36%, 73%, and 48% for roots, stems, and leaves, respectively.

**Table 1.** Means of dry mass (DM; g plant$^{-1}$) of the roots, stems, and leaves and the root/shoot ratio of 28 genotypes of tobacco grown at 2 and 10 mM of nitrogen (N).

| Genotypes | DM [1] (g plant$^{-1}$) | | | | | | Root/Shoot Ratio [2] | |
| | Roots | | Stems | | Leaves | | | |
| | 2 mM | 10 mM | 2 mM | 10 mM | 2 mM | 10 mM | 2 mM | 10 mM |
|---|---|---|---|---|---|---|---|---|
| BAG 06 | 5.23 [B] | 8.05 [E] | 4.20 [A] | 14.20 [B] | 12.58 [A] | 24.31 [A] | 0.31 [E] | 0.21 [D] |
| BAT 2101 | 6.52 [A] | 10.28 [C] | 1.70 [B] | 9.05 [E] | 10.85 [B] | 21.39 [B] | 0.52 [A] | 0.34 [A] |
| BAT 2301 | 5.99 [A] | 8.51 [D] | 3.77 [A] | 13.00 [C] | 11.39 [B] | 21.75 [B] | 0.39 [C] | 0.25 [C] |
| BAT 3004 | 5.34 [B] | 8.76 [D] | 2.56 [B] | 9.60 [E] | 9.98 [B] | 20.55 [B] | 0.42 [C] | 0.29 [B] |
| BAT 3201 | 6.55 [A] | 9.58 [C] | 2.88 [B] | 10.90 [D] | 10.53 [B] | 23.08 [B] | 0.49 [B] | 0.28 [B] |
| CSC 221 | 6.08 [A] | 9.93 [C] | 3.12 [B] | 11.43 [D] | 10.67 [B] | 20.32 [B] | 0.44 [C] | 0.31 [A] |
| CSC 2305 | 6.07 [A] | 9.29 [C] | 2.82 [B] | 13.60 [B] | 10.23 [B] | 22.83 [B] | 0.47 [B] | 0.25 [C] |
| CSC 2307 | 6.38 [A] | 9.83 [C] | 2.40 [B] | 11.54 [D] | 11.16 [B] | 22.02 [B] | 0.47 [B] | 0.29 [B] |
| CSC 259 | 6.90 [A] | 10.61 [B] | 2.93 [B] | 12.10 [C] | 11.13 [B] | 21.24 [B] | 0.49 [B] | 0.32 [A] |
| CSC 2602 | 6.95 [A] | 9.79 [C] | 2.61 [B] | 10.49 [D] | 10.62 [B] | 22.55 [B] | 0.53 [A] | 0.30 [B] |
| CSC 302 | 6.33 [A] | 11.42 [A] | 2.87 [B] | 12.48 [C] | 10.51 [B] | 22.56 [B] | 0.48 [B] | 0.32 [A] |
| CSC 3702 | 6.82 [A] | 11.88 [A] | 2.17 [B] | 11.13 [D] | 11.69 [B] | 23.09 [B] | 0.49 [B] | 0.35 [A] |
| CSC 3703 | 6.14 [A] | 10.92 [B] | 2.25 [B] | 9.45 [E] | 11.22 [B] | 25.53 [A] | 0.46 [B] | 0.31 [A] |
| CSC 416 | 5.17 [B] | 7.71 [E] | 3.76 [A] | 12.18 [C] | 13.15 [A] | 25.12 [A] | 0.30 [E] | 0.20 [D] |
| CSC 4303 | 5.98 [A] | 8.94 [D] | 4.86 [A] | 12.38 [C] | 14.10 [A] | 26.12 [A] | 0.31 [E] | 0.23 [C] |
| CSC 4304 | 5.64 [B] | 8.76 [D] | 3.92 [A] | 13.73 [B] | 14.50 [A] | 26.99 [A] | 0.31 [E] | 0.22 [D] |

**Table 1.** *Cont.*

| Genotypes | DM [1] (g plant$^{-1}$) | | | | | | Root/Shoot Ratio [2] | |
|---|---|---|---|---|---|---|---|---|
| | Roots | | Stems | | Leaves | | | |
| | 2 mM | 10 mM | 2 mM | 10 mM | 2 mM | 10 mM | 2 mM | 10 mM |
| CSC 439 | 5.01 [B] | 8.62 [D] | 3.96 [A] | 13.38 [B] | 13.15 [A] | 24.52 [A] | 0.29 [E] | 0.23 [C] |
| CSC 444 | 5.91 [A] | 8.71 [D] | 3.82 [A] | 12.09 [C] | 15.13 [A] | 24.97 [A] | 0.31 [E] | 0.23 [C] |
| CSC 447 | 5.48 [B] | 7.99 [E] | 3.89 [A] | 13.23 [C] | 14.49 [A] | 25.27 [A] | 0.30 [E] | 0.21 [D] |
| CSC 4501 | 5.43 [B] | 9.43 [C] | 4.02 [A] | 14.76 [A] | 14.52 [A] | 24.89 [A] | 0.29 [E] | 0.24 [C] |
| CSC 4703 | 6.12 [A] | 9.30 [C] | 2.92 [B] | 11.89 [C] | 14.90 [A] | 21.89 [B] | 0.34 [D] | 0.27 [B] |
| CSC 4704 | 5.51 [B] | 8.21 [D] | 3.05 [B] | 10.45 [D] | 14.56 [A] | 24.84 [A] | 0.31 [E] | 0.24 [C] |
| CSC 4707 | 5.36 [B] | 7.10 [E] | 4.71 [A] | 15.64 [A] | 13.47 [A] | 27.66 [A] | 0.29 [E] | 0.16 [E] |
| CSC 497 | 4.84 [B] | 8.43 [D] | 4.23 [A] | 13.96 [B] | 13.19 [A] | 25.52 [A] | 0.28 [E] | 0.22 [D] |
| CSC 500 | 5.37 [B] | 7.56 [E] | 3.17 [B] | 10.50 [D] | 14.13 [A] | 20.40 [B] | 0.31 [E] | 0.24 [C] |
| Dark O.S. | 4.09 [B] | 9.43 [C] | 2.83 [B] | 12.13 [C] | 8.33 [B] | 21.51 [B] | 0.36 [D] | 0.28 [B] |
| HB 4488P | 5.85 [A] | 8.33 [D] | 3.29 [B] | 12.71 [C] | 10.72 [B] | 24.33 [A] | 0.42 [C] | 0.22 [C] |
| New cultivar | 5.77 [A] | 9.01 [D] | 2.62 [B] | 9.91 [E] | 14.32 [A] | 26.11 [A] | 0.34 [D] | 0.25 [C] |
| Average | 5.82 | 9.16 | 3.26 | 12.07 | 12.33 | 23.62 | 0.38 | 0.26 |

Means followed by the same letter do not differ significantly ($p > 0.05$) between genotypes by the Scott–Knott test. [1] Higher means were detected for 10 mM for all genotypes assessed. [2] Higher means were detected for 2 mM for all genotypes assessed.

There was a significant variation in DM production of the different organs studied among genotypes. The lowest root DM was 4.09 g plant$^{-1}$ (Dark O.S.) and the highest was 6.95 (CSC 2602) for the N-deficient treatment; otherwise, 7.10 g plant$^{-1}$ was produced by CSC 4707 and 11.88 by CSC 3702 under adequate N supply. For stem DM at N deficiency, it ranged from 1.70 (BAT 2101) to 4.86 g plant$^{-1}$ (CSC 4303), and 9.05 to 15.64 g plant$^{-1}$ for BAT 2101 and CSC 4707, respectively, under adequate N supply. Regarding leaf DM, Dark O.S. and CSC 444 produced the lowest and highest DM, 8.33 and 15.13 g plant$^{-1}$, respectively, under N deficiency; on the other hand, 20.32 g plant$^{-1}$ was the lowest DM (CSC 221) and 27.66 the highest (CSC 4707). The increase of DM provided by an adequate N supply varied from 32% (CSC 4704) to 131% (Dark O.S.) for roots, 155% (CSC 4303) to 432% (BAT 2101) for stems, and 44% (CSC 500) to 158% (Dark O.S.) for leaves (Table 1). This highlights the importance of N for yield in tobacco production.

Regardless of the organ, higher N contents were detected for plants grown under adequate N supply for all genotypes (Table 2). Among genotypes, the N content varied from 16.66 (BAT 2101) to 20.74 mg g$^{-1}$ (CSC 302) for roots, 13.59 (CSC 497) to 20.94 mg g$^{-1}$ (BAT 2101) for stems, and 13.06 (CSC 4703) to 24.45 mg g$^{-1}$ (Dark O.S.) for leaves when plants were grown in N-deficient solution. Under adequate N supply, the N content ranged from 26.80 (BAT 3201) to 34.54 mg g$^{-1}$ (CSC 4707) in roots, 22.61 (CSC 444) to 28.64 mg g$^{-1}$ (CSC 4704) for stems, and 27.82 (CSC 4707) to 35.81 mg g$^{-1}$ (BAT 3004) for leaves.

**Table 2.** Means of N content (mg g$^{-1}$) in the roots, stems, and leaves of 28 genotypes of tobacco grown at 2 and 10 mM of N.

| Genotypes | N content (mg g$^{-1}$) plant$^{-1}$ | | | | | |
| | Roots | | Stems | | Leaves | |
| | 2 mM | 10 mM | 2 mM | 10 mM | 2 mM | 10 mM |
|---|---|---|---|---|---|---|
| BAG 06 | 17.37 [A] | 30.31 [B] | 15.28 [C] | 23.77 [B] | 16.28 [B] | 30.68 [B] |
| BAT 2101 | 16.66 [A] | 29.65 [C] | 20.94 [A] | 28.00 [A] | 16.75 [B] | 33.03 [A] |
| BAT 2301 | 18.17 [A] | 27.14 [D] | 14.87 [C] | 24.50 [B] | 16.33 [B] | 32.85 [A] |
| BAT 3004 | 18.10 [A] | 28.67 [C] | 19.03 [B] | 27.04 [A] | 18.83 [B] | 35.81 [A] |
| BAT 3201 | 19.23 [A] | 26.80 [D] | 16.93 [C] | 23.65 [B] | 17.67 [B] | 34.36 [A] |
| CSC 221 | 18.85 [A] | 28.10 [C] | 14.99 [C] | 25.51 [A] | 18.74 [B] | 34.19 [A] |
| CSC 2305 | 19.03 [A] | 29.70 [C] | 15.65 [C] | 24.14 [B] | 19.57 [B] | 34.47 [A] |
| CSC 2307 | 19.23 [A] | 32.42 [B] | 16.77 [C] | 26.11 [A] | 17.82 [B] | 33.04 [A] |
| CSC 259 | 18.60 [A] | 29.22 [C] | 18.40 [B] | 26.92 [A] | 16.62 [B] | 33.06 [A] |
| CSC 2602 | 18.24 [A] | 29.01 [C] | 17.35 [B] | 27.06 [A] | 17.41 [B] | 33.85 [A] |
| CSC 302 | 20.74 [A] | 28.97 [C] | 18.74 [B] | 23.07 [B] | 17.72 [B] | 32.16 [A] |
| CSC 3702 | 20.23 [A] | 26.99 [D] | 20.38 [A] | 25.71 [A] | 14.94 [C] | 30.75 [B] |
| CSC 3703 | 19.84 [A] | 28.77 [C] | 18.37 [B] | 26.39 [A] | 17.05 [B] | 29.36 [B] |
| CSC 416 | 20.36 [A] | 32.15 [B] | 16.03 [C] | 24.12 [B] | 14.74 [C] | 29.00 [B] |
| CSC 4303 | 19.79 [A] | 31.64 [B] | 14.63 [C] | 24.12 [B] | 13.50 [C] | 29.92 [B] |
| CSC 4304 | 19.38 [A] | 31.54 [B] | 16.08 [C] | 23.02 [B] | 13.18 [C] | 27.96 [B] |
| CSC 439 | 19.23 [A] | 31.01 [B] | 17.12 [B] | 24.06 [B] | 15.18 [C] | 31.21 [B] |
| CSC 444 | 18.54 [A] | 30.42 [B] | 14.81 [C] | 22.61 [B] | 13.41 [C] | 31.32 [B] |
| CSC 447 | 19.43 [A] | 31.77 [B] | 14.91 [C] | 23.87 [B] | 13.88 [C] | 31.18 [B] |
| CSC 4501 | 19.36 [A] | 28.62 [C] | 15.40 [C] | 23.54 [B] | 13.69 [C] | 28.49 [B] |
| CSC 4703 | 20.03 [A] | 31.71 [B] | 18.29 [B] | 26.63 [A] | 13.06 [C] | 31.98 [A] |
| CSC 4704 | 18.24 [A] | 31.03 [B] | 16.16 [C] | 28.64 [A] | 13.79 [C] | 30.99 [B] |
| CSC 4707 | 19.54 [A] | 34.54 [A] | 15.53 [C] | 22.75 [B] | 13.42 [C] | 27.82 [B] |
| CSC 497 | 18.84 [A] | 28.96 [C] | 13.59 [C] | 22.70 [B] | 16.39 [B] | 30.51 [B] |
| CSC 500 | 18.19 [A] | 29.27 [C] | 14.99 [C] | 25.85 [A] | 14.48 [C] | 34.95 [A] |
| Dark O.S. | 19.03 [A] | 27.40 [D] | 17.56 [B] | 23.98 [B] | 24.45 [A] | 32.60 [A] |
| HB 4488P | 18.25 [A] | 29.15 [C] | 16.03 [C] | 24.09 [B] | 17.86 [B] | 32.86 [A] |
| New cultivar | 19.81 [A] | 26.89 [D] | 15.67 [C] | 25.01 [B] | 14.32 [C] | 31.75 [A] |
| Average | 19.01 | 29.71 | 16.59 | 24.89 | 16.11 | 31.79 |

Means followed by the same letters do not differ significantly ($p > 0.05$) between genotypes by the Scott–Knott test. Higher means were detected in all organs for 10 mM for all genotypes assessed.

The highest N accumulation occurred for all genotypes regardless of the plant organ under adequate N supply (Table 3). N accumulation ranged from 76 (Dark O.S.) to 138 mg N (CSC 3702) in roots, 35.55 (BAT 2101) to 73.22 mg N (CSC 4707) in stems, and 174.71 (CSC 3702) to 212.85 mg N (CSC 497) for leaves when plants were grown under N deficiency. On the other hand, 221.76 (CSC 500) and 330.20 mg N (CSC 302) were the lowest and highest N accumulations in roots whereas, in the stem, this ranged from 246.82 (CSC 3703) to 355.90 mg N (CSC 4707), and from 689.64 (CSC 221) to 827.05 mg N (New cultivar) in the leaves under adequate N supply. Among genotypes, this represents a variation of 81%, 106%, and 22% for roots, stems, and leaves, respectively, under N deficiency, and 49%, 44%, and 20% for roots, stems, and leaves, respectively, when plants were grown in N-sufficient solution. On average, an adequate N supply caused an increase of N accumulation of 145%, 461%, and 284% for roots, stems, and leaves, respectively. The order of N accumulation was stems < roots < leaves for N deficiency, and roots < stems < leaves for adequate N supply on average.

**Table 3.** Means of N accumulation (mg) in the roots, stems, and leaves for 28 tobacco genotypes grown at 2 and 10 mM of N.

| Genotypes | N accumulation (mg) plant$^{-1}$ | | | | | |
|---|---|---|---|---|---|---|
| | Roots | | Stems | | Leaves | |
| | 2 mM | 10 Mm | 2 mM | 10 mM | 2 mM | 10 Mm |
| BAG 06 | 91.02 [B] | 244.36 [D] | 64.22 [A] | 337.34 [A] | 202.71 [A] | 745.81 [B] |
| BAT 2101 | 108.54 [B] | 305.10 [A] | 35.55 [A] | 253.09 [D] | 181.81 [A] | 704.53 [C] |
| BAT 2301 | 108.40 [B] | 230.80 [D] | 55.67 [A] | 317.75 [B] | 185.57 [A] | 710.01 [C] |
| BAT 3004 | 96.48 [B] | 251.36 [D] | 48.58 [A] | 259.20 [D] | 187.65 [A] | 734.77 [C] |
| BAT 3201 | 125.90 [A] | 257.22 [C] | 48.84 [A] | 256.15 [D] | 186.08 [A] | 787.80 [A] |
| CSC 221 | 114.50 [A] | 280.79 [B] | 46.79 [A] | 290.56 [C] | 198.90 [A] | 689.64 [C] |
| CSC 2305 | 115.15 [A] | 275.03 [C] | 43.75 [A] | 327.36 [B] | 198.65 [A] | 785.22 [A] |
| CSC 2307 | 122.23 [A] | 318.42 [A] | 40.23 [A] | 300.60 [B] | 198.41 [A] | 724.76 [C] |
| CSC 259 | 128.67 [A] | 309.69 [A] | 53.85 [A] | 324.82 [B] | 184.77 [A] | 700.05 [C] |
| CSC 2602 | 126.76 [A] | 283.75 [B] | 45.22 [A] | 282.42 [C] | 184.44 [A] | 760.98 [B] |
| CSC 302 | 131.27 [A] | 330.20 [A] | 53.85 [A] | 288.30 [C] | 182.83 [A] | 725.17 [C] |
| CSC 3702 | 138.00 [A] | 320.49 [A] | 44.09 [A] | 285.25 [C] | 174.71 [A] | 708.08 [C] |
| CSC 3703 | 121.72 [A] | 313.82 [A] | 40.81 [A] | 246.82 [D] | 190.20 [A] | 746.71 [B] |
| CSC 416 | 105.18 [B] | 247.94 [D] | 60.11 [A] | 291.35 [C] | 193.12 [A] | 726.13 [C] |
| CSC 4303 | 117.45 [A] | 281.69 [B] | 71.23 [A] | 296.06 [C] | 189.63 [A] | 769.56 [B] |
| CSC 4304 | 109.27 [B] | 275.71 [C] | 63.20 [A] | 316.20 [B] | 190.96 [A] | 750.72 [B] |
| CSC 439 | 96.38 [B] | 267.06 [C] | 68.12 [A] | 321.74 [B] | 199.15 [A] | 758.15 [B] |
| CSC 444 | 109.29 [B] | 264.21 [C] | 56.52 [A] | 272.95 [C] | 202.64 [A] | 779.41 [A] |
| CSC 447 | 106.33 [B] | 253.09 [D] | 58.07 [A] | 315.59 [B] | 201.10 [A] | 776.71 [A] |
| CSC 4501 | 104.96 [B] | 271.13 [C] | 61.90 [A] | 347.38 [A] | 198.28 [A] | 706.71 [C] |
| CSC 4703 | 122.34 [A] | 294.78 [B] | 53.25 [A] | 315.92 [B] | 194.22 [A] | 699.40 [C] |
| CSC 4704 | 100.47 [B] | 254.88 [C] | 49.39 [A] | 286.49 [C] | 200.89 [A] | 756.97 [B] |
| CSC 4707 | 104.71 [B] | 244.91 [D] | 73.22 [A] | 355.90 [A] | 180.71 [A] | 767.62 [B] |
| CSC 497 | 90.63 [B] | 244.73 [D] | 57.61 [A] | 315.74 [B] | 212.85 [A] | 762.15 [B] |
| CSC 500 | 97.55 [B] | 221.76 [D] | 47.60 [A] | 271.01 [C] | 204.69 [A] | 711.63 [C] |
| Dark O.S. | 76.00 [B] | 257.85 [C] | 48.70 [A] | 290.90 [C] | 199.37 [A] | 698.61 [C] |
| HB 4488P | 106.69 [B] | 242.99 [D] | 52.90 [A] | 306.16 [B] | 191.37 [A] | 799.77 [A] |
| New cultivar | 114.25 [A] | 242.07 [D] | 39.99 [A] | 247.36 [D] | 203.89 [A] | 827.05 [A] |
| Average | 110.36 | 270.92 | 52.98 | 297.16 | 193.56 | 743.36 |

Means followed by the same capital letters do not differ significantly ($p > 0.05$) between genotypes by the Scott–Knott test. Regardless of the organ assessed, higher means were detected for 10 mM for all genotypes assessed.

## 3.2. N Use-Efficiency Traits

The NUE, i.e., the amount of plant DM produced per unit of N, was higher in plants under N deficiency, regardless of the genotype (Table 4). NUE ranged from 46.94 (Dark O.S.) to 67.50 g DM g$^{-1}$ N (CSC 444) under N deficiency, and from 31.24 (BAT 3004) to 37.02 g DM g$^{-1}$ N (CSC 4501) under adequate N supply. The physiological N use efficiency (PNUE), i.e., plant DM per N concentration, varied between 0.74 (Dark O.S.) to 1.68 g$^2$ DM mg$^{-1}$ N (CSC 444) for plants subjected to N deficiency, but ranged from 1.21 (BAT 3004) to 1.86 g$^2$ DM mg$^{-1}$ N (CSC 4707) under adequate N supply. When the amount of N in the leaves per unit of N in the whole plant was calculated, the NHI ranged from 48.98% (CSC 3702) to 61.66% (Dark O.S.) for plants grown under N deficiency, and from 52.50% (CSC 259) to 62.78% (New cultivar) under adequate N supply. The NUpE, i.e., the amount of N in the whole plant by root mass, was between 50.02 (BAT 2101) to 82.53 mg N g$^{-1}$ roots (Dark O.S.) under N deficiency, and from 110.65 (CSC 3702) to 194.02 mg N g$^{-1}$ roots (CSC 4707) under adequate N supply. NUpE was higher for plants grown under adequate N supply regardless of genotype, which was reflected in the lower root/shoot ratio for plants conducted at the aforementioned N concentration (Table 1). In respect to the RENU, in terms of the ratio of plant DM under deficient and at adequate N supplies, 35.16 (Dark O.S.) and 59.17% (CSC 500) were the lowest and the highest values, respectively.

**Table 4.** N utilization efficiency (NUE; g DM g$^{-1}$ N), physiological N use efficiency (PNUE; g$^2$ DM mg$^{-1}$ N), N harvest index (NHI; %), N uptake efficiency (NUpE; mg N g$^{-1}$ roots), and relative efficiency of N use (RENU; %) for 28 tobacco cultivars grown at 2 and 10 mM of N.

| Genotypes | NUE [1] (g DM g$^{-1}$ N) | | PNUE (g$^2$ DM mg$^{-1}$ N) | | NHI (%) | | NUpE [2] (mg N g$^{-1}$ roots) | | RENU (%) |
|---|---|---|---|---|---|---|---|---|---|
| | 2 mM | 10 mM | 2 mM | 10 mM | 2 mM | 10 mM | 2 mM | 10 mM | |
| BAG 06 | 61.51 [B] | 35.08 [A] | 1.36 [Ab] | 1.63 [Aa] | 56.64 [Ba] | 56.16 [Ba] | 69.04 [A] | 165.42 [B] | 47.23 [C] |
| BAT 2101 | 58.51 [C] | 32.24 [A] | 1.12 [Ba] | 1.31 [Ba] | 55.78 [Ba] | 55.78 [Ba] | 50.02 [B] | 122.80 [E] | 46.92 [C] |
| BAT 2301 | 60.47 [B] | 34.33 [A] | 1.28 [Ba] | 1.49 [Ba] | 53.07 [Ca] | 56.39 [Ba] | 58.67 [B] | 148.44 [C] | 48.97 [C] |
| BAT 3004 | 53.75 [C] | 31.24 [A] | 0.96 [Ba] | 1.21 [Ba] | 56.39 [Ba] | 59.02 [Aa] | 62.56 [B] | 142.37 [D] | 45.97 [C] |
| BAT 3201 | 55.33 [C] | 33.44 [A] | 1.11 [Bb] | 1.46 [Ba] | 51.58 [Cb] | 60.61 [Aa] | 55.10 [B] | 136.23 [D] | 46.06 [C] |
| CSC 221 | 55.17 [C] | 33.00 [A] | 1.10 [Bb] | 1.38 [Ba] | 55.22 [Ba] | 54.87 [Ba] | 59.36 [B] | 130.16 [D] | 47.90 [C] |
| CSC 2305 | 53.40 [C] | 32.92 [A] | 1.03 [Bb] | 1.51 [Ba] | 55.50 [Ba] | 56.60 [Ba] | 59.15 [B] | 149.83 [C] | 41.80 [D] |
| CSC 2307 | 55.28 [C] | 32.25 [A] | 1.11 [Bb] | 1.40 [Ba] | 55.00 [Ba] | 53.97 [Ba] | 56.94 [B] | 136.86 [D] | 46.02 [C] |
| CSC 259 | 57.08 [C] | 32.89 [A] | 1.20 [Ba] | 1.45 [Ba] | 50.32 [Ca] | 52.50 [Ba] | 53.31 [B] | 126.83 [E] | 47.77 [C] |
| CSC 2602 | 56.59 [C] | 32.23 [A] | 1.14 [Ba] | 1.38 [Ba] | 51.76 [Cb] | 57.32 [Aa] | 51.31 [B] | 135.77 [D] | 47.25 [C] |
| CSC 302 | 53.63 [C] | 34.60 [A] | 1.06 [Bb] | 1.61 [Aa] | 49.74 [Cb] | 53.98 [Ba] | 58.15 [B] | 117.95 [E] | 42.41 [D] |
| CSC 3702 | 58.00 [C] | 35.07 [A] | 1.20 [Bb] | 1.62 [Aa] | 48.98 [Cb] | 53.88 [Ba] | 52.38 [B] | 110.65 [E] | 44.97 [C] |
| CSC 3703 | 55.54 [C] | 35.10 [A] | 1.09 [Bb] | 1.61 [Aa] | 53.98 [Ba] | 57.08 [Aa] | 57.50 [B] | 120.10 [E] | 42.78 [D] |
| CSC 416 | 61.53 [B] | 35.56 [A] | 1.36 [Aa] | 1.60 [Aa] | 53.94 [Ba] | 57.40 [Aa] | 69.36 [A] | 165.44 [B] | 49.08 [C] |
| CSC 4303 | 65.90 [A] | 35.09 [A] | 1.65 [Aa] | 1.68 [Aa] | 50.16 [Cb] | 57.16 [Aa] | 63.66 [A] | 153.09 [C] | 53.02 [B] |
| CSC 4304 | 66.23 [A] | 36.84 [A] | 1.59 [Aa] | 1.83 [Aa] | 52.55 [Ca] | 55.93 [Ba] | 64.68 [A] | 153.62 [C] | 48.73 [C] |
| CSC 439 | 60.87 [B] | 34.51 [A] | 1.35 [Aa] | 1.61 [Aa] | 54.85 [Ba] | 56.27 [Ba] | 72.59 [A] | 156.69 [B] | 47.64 [C] |
| CSC 444 | 67.50 [A] | 34.74 [A] | 1.68 [Aa] | 1.59 [Aa] | 55.01 [Bb] | 59.19 [Aa] | 62.59 [B] | 151.93 [C] | 54.43 [B] |
| CSC 447 | 65.30 [A] | 34.52 [A] | 1.56 [Aa] | 1.62 [Aa] | 55.02 [Ba] | 57.70 [Aa] | 66.78 [A] | 168.58 [B] | 51.63 [B] |
| CSC 4501 | 65.62 [A] | 37.02 [A] | 1.57 [Aa] | 1.82 [Aa] | 54.28 [Ba] | 53.38 [Ba] | 67.41 [A] | 141.32 [D] | 48.85 [C] |
| CSC 4703 | 64.67 [A] | 32.87 [A] | 1.55 [Aa] | 1.42 [Ba] | 52.51 [Ca] | 53.38 [Ba] | 60.49 [B] | 141.50 [D] | 55.54 [A] |
| CSC 4704 | 65.97 [A] | 33.42 [A] | 1.52 [Aa] | 1.47 [Ba] | 57.27 [Ba] | 58.30 [Aa] | 63.67 [A] | 158.31 [B] | 53.94 [B] |
| CSC 4707 | 65.58 [A] | 36.85 [A] | 1.54 [Ab] | 1.86 [Aa] | 50.35 [Cb] | 56.12 [Ba] | 66.93 [A] | 194.02 [A] | 46.68 [C] |
| CSC 497 | 61.52 [B] | 36.14 [A] | 1.38 [Ab] | 1.74 [Aa] | 59.00 [Aa] | 57.61 [Aa] | 75.18 [A] | 158.70 [B] | 46.51 [C] |
| CSC 500 | 64.84 [A] | 31.90 [A] | 1.47 [Aa] | 1.23 [Ba] | 58.49 [Aa] | 59.11 [Aa] | 65.27 [A] | 160.28 [B] | 59.17 [A] |
| Dark O.S. | 46.94 [D] | 34.55 [A] | 0.74 [Bb] | 1.49 [Ba] | 61.66 [Aa] | 56.02 [Bb] | 82.53 [A] | 133.76 [D] | 35.16 [E] |
| HB 4488P | 56.60 [C] | 33.66 [A] | 1.12 [Bb] | 1.53 [Ba] | 54.54 [Bb] | 59.27 [Aa] | 60.01 [B] | 162.16 [B] | 43.77 [D] |
| New cultivar | 63.41 [B] | 34.19 [A] | 1.45 [Aa] | 1.54 [Ba] | 56.96 [Bb] | 62.78 [Aa] | 62.07 [B] | 146.31 [C] | 50.37 [B] |
| Average | 59.88 | 34.15 | 1.30 | 1.54 | 54.31 | 56.71 | 62.38 | 146.04 | 47.88 |

Means followed by the same capital letters do not differ significantly ($p > 0.05$) between genotypes, and means followed by the same lowercase letters do not differ significantly ($p > 0.05$) between N concentrations. [1] Higher means for 2 mM for all genotypes assessed. [2] Higher means for 10 mM for all genotypes assessed.

### 3.3. Grouping Tobacco Genotypes According to their Efficiency and Responsiveness to N Supply

Tobacco genotypes were classified into four different groups: efficient and responsive (ER), efficient and non-responsive (ENR), inefficient and responsive (NER), and inefficient and non-responsive (NENR) to N supply. For this, the NUEf and the plant DM under N deficiency for each genotype were used to plot the y and x axis, respectively. The average NUEf and plant DM for all genotypes were calculated, and are represented by the internal lines of the chart in Figure 1. After plotting all genotypes in the chart, each genotype was allocated to one of four quadrants (groups). The genotypes on the right of the chart are efficient, but those on the left are inefficient in N use. Furthermore, genotypes above the horizontal line are responsive, and below are non-responsive to N supply. Interestingly, the 28 genotypes assessed were equally distributed into the four groups. The genotypes belonging to the varietal groups Virginia and Maryland were efficient, while the genotypes of the Burley, Comum, and Dark groups were inefficient in N use. CSC 500 and New cultivar (both Maryland) were non-responsive, while Virginia, Comum, and Burley groups had genotypes with variations in responsiveness to N supply depending on the genotype studied.

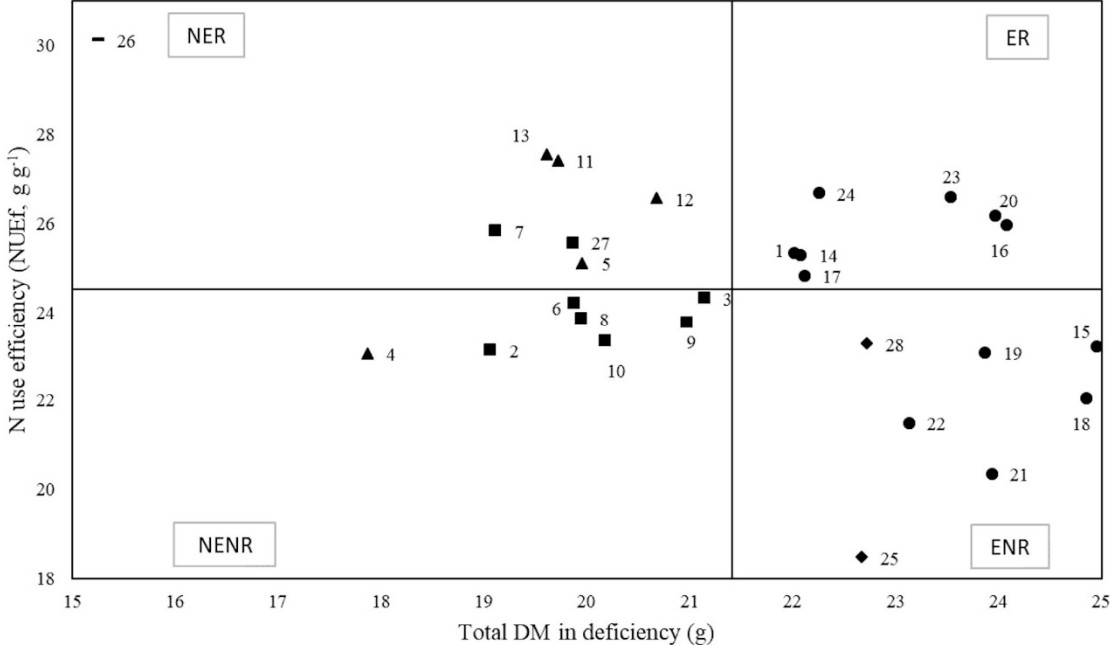

**Figure 1.** Classification of the 28 tobacco genotypes in response to the total dry mass (DM) produced under N deficiency (g) and the responsiveness to N supply (NUEf; g g$^{-1}$). Internal lines represent the mean values for the axis of all genotypes. The Virginia, Burley, Comum, Maryland, and Dark varietal groups are represented by circles, squares, triangles, rhombs, and lines, respectively. ER, efficient and responsive; ENR, efficient and non-responsive; NER, inefficient and responsive; NENR, inefficient and non-responsive. For genotypes and their respective numbering, refer to Table S1.

## 4. Discussion

### 4.1. Variation of N Utilization Efficiency (NUE) Traits among Genotypes and N Supplies

By the use of two nutritive solutions with contrasting N concentrations, deficient and sufficient in N, we investigated the response of a wide range of tobacco genotypes in terms of N use. The N concentrations chosen in the present study provided plants with either a deficient or an adequate supply of N. This was visually observed by the yellowish color of older leaves compared to the light green color of younger leaves in plants grown in nutritive solution with 2 mM of N, but darker green color of the whole plants grown in 10 mM of N. Furthermore, a decrease in DM, N content, and N accumulation for roots, stems, and leaves was caused when plants were subjected to N deficiency,

regardless of the genotype. This was reflected in a higher root/shoot DM ratio under N deficiency, which typically occurs in plants subjected to N deficiency [24]. A higher ratio root/shoot DM ratio in higher N rates was reported by Brueck and Senbayram [10] in two tobacco varieties; however, the authors attributed this fact to losses of fine roots during the washing of the peat–perlite substrate used for cultivation. Indeed, Poorter et al. [25] relate the challenge of assessing a reliable mass of roots in plants conducted in pot experiments with substrate. Fan et al. [15] found a decrease of root/shoot ratio with increasing N concentration for one cultivar, but not in another one when supplied solely with N–$NO_3$. However, our data are in accordance with the "functional equilibrium" existing between roots and shoots described by Brouwer [26]. When the limiting factor is a nutrient (e.g., N), more biomass is allocated to the roots in comparison to the shoots [25]. Thus, plants favor the organ (roots) responsible for the uptake of the limiting factor (N) that plants are experimenting in a given situation [27]. It is likely that genotypes with a higher root/shoot ratio under N depletion are more sensitive to N deficiency.

The higher production of DM of shoots compared to roots under adequate N supply is related to the capacity of plants to uptake N from the growth media through the roots, as revealed by the NUpE index, calculated as the amount of N in the plant by the mass of the roots. On average, the NUpE of genotypes grown under adequate N supply was 146.04 mg N $g^{-1}$ roots, while plants under N deficiency showed only 62.38 mg N $g^{-1}$ roots. The NHI is calculated as the N accumulation in leaves by the N in the whole plant. In other words, it represents the amount of N that is allocated to leaves by plants. Thus, it may represent an important index for the tobacco market, as leaves are the commercial product of the crop. The NHI was, on average, quite close among plants grown under N-deficient and N-sufficient supplies: 54.31% and 56.71%, respectively. This means that, regardless of the N nutrition status of tobacco genotypes, the N accumulation in leaves compared to the whole plant remains constant in the range of 50%. Thus, provides valuable information regarding N exported from the crop area in a given growing season [28], which may be helpful for N fertilization management. The higher NHIs suggest that these genotypes are more efficient in transferring N to leaves from root N uptake [15].

The DM produced per unit of N, i.e., the NUE index, was higher for plants grown under N deficiency, regardless of the genotype. A decrease of 3–6-fold between the same N concentrations tested in this experiment was found by Fan et al. [15] for two varieties of tobacco. The authors suggest that the highest concentration supplied plants excessively in N at the growth stages tested. However, this can be explained simply by the well-known law of diminishing returns, which postulates that marginal yield decreases as the level of the limiting factor is raised. As the N rates are increased, NUE is decreased, as demonstrated by Sisson et al. [13], in 12 cultivars of tobacco. In this study, 10 mM was the N concentration of the nutritive solution that adequately fertilized tobacco genotypes, and thus, plants grown in it returned 34 g DM $g^{-1}$ N on average, unlike the 60 g DM $g^{-1}$ N produced by plants under N deficiency. Genotypes with higher NUE produce more biomass per unit of N, and they utilize N from the media more efficiently.

The RENU is related to the total DM produced under N-deficient and N-sufficient supplies. It was shown that under N deficiency, the plant DM decreased from 35% (Dark O.S.) to 59% (CSC 500). As leaves are responsible for the most DM in the plant (Table 1), the importance of N fertilization for tobacco genotypes can be confirmed. However, the decrease in plant DM varied considerably among genotypes. Therefore, the genotypic variation should be considered in N fertilizations by landholders.

### 4.2. Classification of Genotypes: Efficiency and Responsiveness to N Supply of Contrasting Tobacco Genotypes

In this study, we classified the tobacco genotypes according to their efficiency and responsiveness to N supply. This was achieved by the use of plant DM obtained by the growth in an N-deficient nutritive solution, and the NUEf was calculated based on the DM and N accumulation obtained of the plants grown in the contrasting N concentration solutions. The combination of both variables resulted in the separation of genotypes into four groups: ER, ENR, NER, and NENR, being ER the group of genotypes most desirable [28]. We chose a hydroponic condition for this study, as the growth of the

tobacco genotypes in soil (i.e., pot or field trial) could result in an unreliable classification due to the unknown amount of mineral N supplied to plants through organic matter mineralization. In addition, the use of nutritive solutions allows the supply N in concentrations previously established.

The classification performed in this study indicated the Virginia and Maryland varietal groups as efficient in N use, which means they produced plant DM above the average of all of the genotypes assessed. The Burley, Comum, and Dark varietal groups were N-inefficient, that is, they produced below average DM. The responsiveness within varietal groups varied considerably, except for the Maryland and Dark groups. This suggests that even individuals belonging to a same genetic group may have different characteristics in the responsiveness to N supply. For example, CSC 497 and CSC 4703, both belonging to the Virginia group, differ considerably in responsiveness, the first being responsive and the latter non-responsive to N. The data of this study may be interpreted from two interesting and practical points of view. First, the responsiveness to N fertilization in the cultivation of a given genotype may be different from another one, even if both belong to a same varietal group (e.g., CSC 3703 and BAT 3004, CSC 4707 and CSC 4703, and the others in Figure 1). Thus, N fertilizer management should preferentially be adjusted according to each genotype, rather than considering the same N demand for different genotypes of a same group. Second, the characteristics of efficiency and responsiveness of each genotype allow the choice of suitable genotypes according to soil N level or affordability of N fertilizers by landholders. However, this is dependent on the edaphoclimatic condition requirements and other desirable plant features such as disease resistance, yield, planting season, or cropping cycle.

The present study provides valuable data to allow further deep investigation of the genotypes assessed. This is important for future breeding targeting the improvement of the N metabolism, and consequently, a better N use efficiency for tobacco genotypes. Few studies have considered the use of numerous tobacco genotypes studied concomitantly to characterize N-use efficiency traits (e.g., [14,15]). This is likely due to the time-consuming process involved in plant growth and analysis. However, the screening of contrasting genotypes is a valuable tool to characterize genotypes regarding their N use and selection. For example, modern cultivars showed generally higher NUE, a 50% higher yield compared to older ones in the work of Sisson et al. [13], which compared 12 cultivars. According to the authors, as much is known about genetic diversity coupled with NUE, it is possible to make future improvements in the cultivars. Among six tobacco cultivars, Ruiz et al. [14] selected the one based on the highest $NO_3$ reductase activity and the lowest foliar $NO_3$ concentration. The authors successfully obtained plants with higher NUE traits by grafting cultivars with less NUE traits to the best cultivar. However, in our view, a better approach to increase NUE in tobacco crops is firstly to characterize as many genotypes as possible through simple methods. This could then comprise a greater number of genotypes spanning a wide range of mechanisms related to NUE traits [9]. In the present study, through simple data of DM and N content of tobacco genotype organs, we characterized and classified 28 genotypes, which were first thought to have contrasting responses to different N supplies. Thus, our data represent interesting information to further deepen investigation regarding the internal mechanisms related to NUE in tobacco crops. In this way, the contrasting genotypes can be subjected to physiological and molecular studies, providing clues about the physiological traits better related to NUE in those tobacco lines, contributing to tobacco breeding programs. The knowledge of more genes related to NUE may contribute to enlarge the breeding approaches aiming to obtain lower TSNAs in tobacco [19].

The use of efficient genotypes is a strategy to improve NUE in a given crop [28]. In tobacco crops, there are serious concerns regarding the management of N fertilization. First, N rates vary widely among tobacco types, from 56 to 308 kg N ha$^{-1}$ [29,30]. Second, high N rates may increase N–$NO_3$ content in the leaves of tobacco [31], which can be further transformed to N–$NO_2$. Then, N–$NO_2$ reacts with precursor alkaloids, forming TSNAs, widely known to be carcinogenic compounds that affect the health of consumers. Third, nicotine, which is the main alkaloid in tobacco, is an addictive substance that causes harmful effects on human health. Nicotine content in leaves increases with N supply [32,33];

therefore, genotypes with a lower demand for N will probably have a lower concentration of nicotine. Thus, taking into account the reaction of nicotine with $N-NO_2$ to form TSNAs [18], genotypes with higher NUE would be desirable as they would be less harmful to healthy humans because of a lower TSNA content [30]. Finally, if $N-NO_3$ undesirably accumulates in leaves, it is likely that a step in N metabolism is impaired [34], which may be caused by excessive use of N fertilizers for any genotype, but such an effect is probably raised in low-efficiency genotypes.

## 5. Conclusions

The 28 tobacco genotypes studied vary in their efficiency and responsiveness to N supply. The genotypes with the highest efficiency for N supply belong to the Virginia and Maryland varietal groups. The other tobacco genotypes belonging to the Burley, Comum and Dark groups are inefficient in terms of N supply. The tobacco genotypes' responsiveness to N supply varies within each of the varietal groups assessed, except for Maryland.

The discrimination of the contrasting tobacco genotypes regarding their NUE in this study may contribute to tobacco-breeding programs. Furthermore, it helps to choose the most suitable genotype depending on soil N availability or affordability of N fertilizers for landholders worldwide. This can contribute to the improvement of the N use in tobacco crops, while preserving natural resources.

**Supplementary Materials:** The following are available online at http://www.mdpi.com/2073-4395/10/4/572/s1: Table S1. Tobacco genotypes and their respective varietal groups used in the experiment; Table S2. Stock solutions and amounts used to prepare the nutritive solution with 2 mM of N (deficiency) and 10 mM of N (adequate); Figure S1. Images showing the visual aspect of tobacco genotypes when subjected to N deficiency (2 mM, pots on the left) or adequate N supply (10 mM, pots on the right).

**Author Contributions:** Conceptualization, A.B.A., D.R.G., V.F., and F.S.C.; Funding acquisition, D.R.G. and V.F.; Investigation, A.B.A., C.S.d.C.S., G.P.F., K.R.D.S., and W.F.T.C.; Methodology, A.B.A., D.R.G., V.F., and F.S.C.; Project administration, A.B.A., D.R.G., V.F., and F.S.C.; Supervision, D.R.G., V.F., and F.S.C.; Writing—original draft, A.B.A.; Writing—review and editing, A.B.A., D.R.G., V.F., F.S.C., and K.R.D.S. All authors have read and agreed to the published version of the manuscript.

**Funding:** This research was funded by Souza Cruz S/A, grant number 113/2017.

**Acknowledgments:** The authors are thankful to Coordination for the Improvement of Higher Education Personnel (Capes) for providing scholarships to the first and seventh authors (financial code 001). They also thank the State of Minas Gerais Research Foundation (Fapemig) and the National Council for Scientific and Technological Development (CNPq) for providing additional financial support, and Larissa B. Barbosa for valuable help with the experiment. Finally, we are thankful for two anonymous reviewers for their valuable suggestions.

**Conflicts of Interest:** The authors declare no conflicts of interest.

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
