# Peer review of "Genotypic Variation in Nitrogen Use-Efficiency Traits of 28 Tobacco Genotypes"

_agronomy, doi:10.3390/agronomy10040572_

Round 1

Reviewer 1 Report

Andrade et al. surveyed tobacco NUE with 28 genotypes and presented various responsiveness of varietal groups to N supply. NUE is increasingly concerned in the modern agriculture under climate changing. How to improve and optimize NUE has attracted much efforts and interests for the obvious reasons. The strong association between N and TSNA add extra practical importance in manipulation of NUE in tobacco, and this is also why I accepted to review this manuscript.

I anticipated a genome-wide association study with a core collection of tobacco germplasm, and the contributing loci to NUE were thoroughly revealed and localized onto tobacco chromosomes. I understand the academic background of authors determines the perspective of research conduction. Still, I strongly suggest the authors may collaborate with geneticists in the follow-up studies to identify QTLs of NUE in tobacco.

The gene controlling NUE in tobacco was already reported by Edwards et al. (2017). I wish this paper can be mentioned or cited in the manuscript, although this may be beyond the authors’ expertise.

An extensive editing of English language is necessary for this manuscript. The grammars and writing skills make the manuscript very challenging to read smoothly. Below are some, but not all, of rough sentences or paragraphs:

  1. Line 50, “That is understandable due to the time consuming in the…”
  2. Line 52, “…to embrace the highest number of genotypes as possible, allowing to know the NUE specifically to each genotype …”
  3. Line 55, “…as suggested by [9].” I don’t think it’s an appropriated way use citation.
  4. Line 58, “The study of genotypes under controlled conditions is faster, allows more genotypes to be investigated, is independent of pathogens and weather and has minimal environmental variability.”
  5. Line 62, “Then, after characterization of the NUE by plants, the choice of the suitable genotype according to its N 62 demand could be performed according to the area to be cultivated.”
  6. Line 63, “For example, to increase NUE in tobacco, [11] simply suggest the selection of the desired genotype.” I don’t think it’s an appropriated way use citation, and it is commonly observed in the manuscript, such in the part of Materials and Methods.
  7. Line 72, “Again, this is comprehensive since the growth of many genotypes is labor-time, and plant…”

I cannot list them all here, please use language editing service before resubmission of this manuscript.

Author Response

Dear reviewer 1:

We hereby resubmit an improved version of our manuscript (agronomy-730635) based on your comments. We express our thanks for the valuable suggestions, which greatly improved the article.

We carefully took note of the suggestions and made all of the suggested corrections, as specified below. The English was reviewed by experts. We believe we have fully complied with your suggestions.

Point 1: The gene controlling NUE in tobacco was already reported by Edwards et al. (2017). I wish this paper can be mentioned or cited in the manuscript, although this may be beyond the authors’ expertise.

Response 1: We cited Edwards et al. (2017) and this strengthened the value that our manuscript has for geneticists in future studies. Thus, that motivated us to write a paragraph (lns 90-96) talking about the importance of our study, and that improved the introduction section.  

Point 2: An extensive editing of English language is necessary for this manuscript. The grammars and writing skills make the manuscript very challenging to read smoothly.

Response 2: Our manuscript was submitted to a language editing service before resubmission.

Point 3: Line 55, “…as suggested by [9].” I don’t think it’s an appropriated way use citation.

Line 63, “For example, to increase NUE in tobacco, [11] simply suggest the selection of the desired genotype.” I don’t think it’s an appropriated way use citation, and it is commonly observed in the manuscript, such in the part of Materials and Methods.

Response 3: We reviewed the entire manuscript regarding citations in the text and references before resubmission.

Reviewer 2 Report

The manuscript “Genotypic Variation in Nitrogen Use Efficiency Traits of 28 Tobacco Genotypes” describes the process of identifying tobacco cultivars with an increased nitrogen use efficiency in order to improve the selection of resource saving cultivars.

The experimental framework is transparent and comprehensive, data is presented clearly as well as shown in appropriate tables, a meaningful figure was created to classify genotypes, and findings are highlighted and discussed appropriately.

However, the potential scope for practical approaches, e.g. for tobacco production process and/or breeding strategies, need to be strengthen in more detail. It was to be expected that plants grown with less nitrogen have a lower shoot and root biomass compared with plants cultivated under adequate nitrogen supply. Thus, it is important to link your valuable findings/ your cost-efficient selection approach of 28 lines at the same time with the practical background of tobacco production.

After identifying tobacco lines 1, 14, 16, 17, 20, 23 and 24 as promising ones (efficient and responsive; Figure 1), the following issues should be clearly and concisely discussed i) assessment of further (necessary) physiological parameters to characterize promising lines to ensure benefits for production/breeding, ii) feasibility/necessity for tobacco farmers to reduce nutrition/ to cultivate other cultivars than their usual lines in the field, and iii) the effect of nitrogen shortage for the amount of essential nicotine content in leaves versus undesired amount of nitrite (as potential negatively related factor). At the end of the discussion, line 320 ff, you turn briefly to this topic (nitrate/undesired nitrite content), which should be highlighted as important aspect of tobacco production. Besides, do you have distinguished between nitrate and nitrite when assessing data of nitrogen content?
Particularly, the fact that tobacco lines (seedlings/young plants) were grown hydroponically offers wide possibilities to speculate if a seasonal soil or field experiment would result into the same selection of promising genotypes. Of course, basic research is necessary and mandatory, but you should nevertheless refer to this critical issue in your discussion.
Please, consider to summarize your main findings in the conclusion. This would also avoid the given repetition of outcomes at the end of the discussion part.

Besides, it would be necessary to state (with previous experiments or other studies), why you have decided to use 2mM nitrogen as “deficient” and 10mM as “sufficient”. Furthermore, I do see the point to compare “deficiency” with “sufficiency”, but “sufficiency” is a rarely used wording that interferes with the flow of reading. Instead of “sufficiency” I would suggest to use the wording “control conditions”, “sufficient nitrogen supply” or “adequate N supply” (as so in line 159) etc. Furthermore, in line 144 you state that plants under nitrogen deficiency showed a light green color compared with controls. Do you have further data to emphasize this physiological changes, such as fluorescence measurements or at least images of the plants (for supplementary)?

In general, in the context of writing style, I would suggest to use more frequently adjectives instead of long complex descriptions, e.g. instead of “nitrogen use efficiency improvement” (line 33) use “improved nitrogen use efficiency”.  Rephrasing would also be necessary for several other sentences such as e.g. in line 56-57,63-64, 84 (we aimed in this study), and for “in a big picture” which is a very colloquial expression.
In some cases, the main point of sentences were not clear to me, such as e.g. in line 71-73, 80-81, 271-272, 277-278, and 288-290.
In regard to the equations, please add brackets in (6), line 135, to clearly state how NUEf is calculated (point before line calculation). In addition, please clarify in (2) if you refer indeed to square gram (g²= g x g).

With major revisions, including an improved emphasis on the practical implementation of research finding into (potential) production processes and breeding aims, as well as suggested improvements, the manuscript will be of value for the Journal MDPI agronomy.

Author Response

Dear reviewer 2:

We hereby resubmit an improved version of our manuscript (agronomy-730635) based on your comments. We express our thanks for the valuable suggestions, which greatly improved the article.

We carefully took note of the suggestions and made all of the suggested corrections, as specified below. The English was reviewed by experts. We believe we have fully complied with your suggestions.

Point 1: However, the potential scope for practical approaches, e.g. for tobacco production process and/or breeding strategies, need to be strengthen in more detail.

It was to be expected that plants grown with less nitrogen have a lower shoot and root biomass compared with plants cultivated under adequate nitrogen supply.

Thus, it is important to link your valuable findings/ your cost-efficient selection approach of 28 lines at the same time with the practical background of tobacco production.

Response 1: We highlighted the importance of our study regarding practical approaches and that motivated us to improve the introduction section in lns 65-67, 89-96. Thus, the contribution of our study for geneticists to improve the quality of tobacco products, e.g. low N compounds, was also addressed.

Plants grown at N deficient solution had lower root, stem and leaves dry mass, as shown in table 1. However, the ratio between root and shoot dry mass (column root/shoot ratio in table 1) is even higher for plants conducted at N deficiency compared to plants at adequate N supply. This is discussed in lns 266-280.

Due to an increase of 44 to 158% in leaves dry mass when N was adequately supplied, we briefly included this practical importance in the tobacco production in lns 178-9. Another practical backgrounds are in lns 287-288, 292-293, 310-311.

Point 2: After identifying tobacco lines 1, 14, 16, 17, 20, 23 and 24 as promising ones (efficient and responsive; Figure 1), the following issues should be clearly and concisely discussed i) assessment of further (necessary) physiological parameters to characterize promising lines to ensure benefits for production/breeding, ii) feasibility/necessity for tobacco farmers to reduce nutrition/ to cultivate other cultivars than their usual lines in the field, and iii) the effect of nitrogen shortage for the amount of essential nicotine content in leaves versus undesired amount of nitrite (as potential negatively related factor). At the end of the discussion, line 320 ff, you turn briefly to this topic (nitrate/undesired nitrite content), which should be highlighted as important aspect of tobacco production. Besides, do you have distinguished between nitrate and nitrite when assessing data of nitrogen content?

Response 2: i) Our study may support further deep investigations in physiological and molecular studies. Since contrasting genotypes were identified, this may be used to study any physiological trait related to NUE which will be used to improve/develop new genotypes with improved characteristics, such as low N compounds with acceptable yield (lns 359-363). Indeed, we are working in a second trial using the contrasting genotypes for more details in the physiological parameters.

  1. ii) We discussed the importance of knowing the NUE of wide range of tobacco genotypes and the applicability of that by landholders. Initially, we briefly mentioned the importance for farmers in the introduction section (lns 65-67). Later, we discussed the implications of our results in practical way for farmers in lns 330-339.

iii) N effects on nicotine content were discussed at the end of the discussion section (lns 369-376). We linked NUE, NO2, nicotine and TSNA aspects, that we think it enhanced the discussion and provided practical implications for tobacco production.

In this manuscript, the N content was assessed through sulfuric digestion, i.e. N content comprises all forms of N, since the aims of the study and methodology do not require discrimination of the mineral N forms.

Point 3: Particularly, the fact that tobacco lines (seedlings/young plants) were grown hydroponically offers wide possibilities to speculate if a seasonal soil or field experiment would result into the same selection of promising genotypes. Of course, basic research is necessary and mandatory, but you should nevertheless refer to this critical issue in your discussion.

Response 3: Hydroponic conditions is the ideal way for screening studies as it is possible to control the only source of variation (i.e. N) to be studied. Soil provides N through the organic matter mineralization, which adds an additional, undesirable and unknown amount of N to plants. Field conditions add an extra complicating factor which is the soil characteristics variability (i.e. soil fertility, water content, etc.), beyond requiring a higher plot area (plot area in field > pot in greenhouse). We briefly addressed this issue in the introduction section (lns 57-62) and discussion section in lns 318-322. So the results should be questionable as there are many sources of variation in the field.     

Point 4: Please, consider to summarize your main findings in the conclusion. This would also avoid the given repetition of outcomes at the end of the discussion part.

Response 4: The conclusion section was inserted and was rewritten. We considered the mains findings of the work and we mentioned the implications on practical issues in the tobacco production and breeding.

Point 5: Besides, it would be necessary to state (with previous experiments or other studies), why you have decided to use 2mM nitrogen as “deficient” and 10mM as “sufficient”. Furthermore, I do see the point to compare “deficiency” with “sufficiency”, but “sufficiency” is a rarely used wording that interferes with the flow of reading. Instead of “sufficiency” I would suggest to use the wording “control conditions”, “sufficient nitrogen supply” or “adequate N supply” (as so in line 159) etc.

Response 5: We really ran many test trials with different N concentrations with genotypes of Virginia and Burley varietal groups prior to the experiment presented in the manuscript. The visual symptoms presented by the plants in the different treatments ensured us that the concentrations of 2 and 10 mM would safely produce plants deficient and adequately N supplied. This was confirmed by the dry mass and visual symptoms of plants showed in the experiment, and images of some genotypes were included in the supplementary material to show this. We briefly addressed this in lns 120-1. We replaced the word ‘sufficiency’ with ‘adequate N supply’ in the entire manuscript.

Point 6: Furthermore, in line 144 you state that plants under nitrogen deficiency showed a light green color compared with controls. Do you have further data to emphasize this physiological changes, such as fluorescence measurements or at least images of the plants (for supplementary)?

Response 6: We have images showing the visual symptoms of N deficiency and we added images of some genotypes in the supplementary material as suggested (Figure S1). Unfortunately, fluorescence measurements were not used in this study as we don’t have the equipment available.

Point 7: In general, in the context of writing style, I would suggest to use more frequently adjectives instead of long complex descriptions, e.g. instead of “nitrogen use efficiency improvement” (line 33) use “improved nitrogen use efficiency”.  Rephrasing would also be necessary for several other sentences such as e.g. in line 56-57,63-64, 84 (we aimed in this study), and for “in a big picture” which is a very colloquial expression.

In some cases, the main point of sentences were not clear to me, such as e.g. in line 71-73, 80-81, 271-272, 277-278, and 288-290.

Response 7: Those points raised by the reviewer are regarding the writing style, so we submitted the manuscript to a language editing service before resubmission.

Point 8: In regard to the equations, please add brackets in (6), line 135, to clearly state how NUEf is calculated (point before line calculation). In addition, please clarify in (2) if you refer indeed to square gram (g²= g x g).

Response 8: Brackets were added in equation (6). Regarding equation (2), it is really ‘g2 DM mg-1 of N’. That is the result of ‘g of plant dry mass’ divided by ‘N concentration’. So we have ‘[g]/[mg/g]’. That ends in g x g/mg = g2 mg-1.

Round 2

Reviewer 1 Report

The authors addressed all my comments, and the manuscript is well read after extensive editing. I would like to suggest to accept it for publication.

Reviewer 2 Report

Dear authors,

Thank you for your detailed response.
Your manuscript has significantly improved by addressing additional issues such as experimental details as well as references to practical requirements. The English style is excellent, at least in my opinion.

In the present form your manuscript is ready for publication and will be of great interested for readers of Agronomy.